# Association of angiotensin-converting enzyme gene insertion/deletion polymorphisms with risk of hypertension among the Ethiopian population

Tsegaye Adane Birhan[1,2]*, Meseret Derbew Molla[2], Mohamed Abdulkadir[3], Kibur Hunie Tesfa[2]

1 Department of Environmental and Occupational Health and Safety, Institute of Public Health, College of Medicine and Health Sciences, University of Gondar, Gondar, Ethiopia, 2 Department of Biochemistry, School of Medicine, College of Medicine and Health Sciences, University of Gondar, Gondar, Ethiopia, 3 Department of Internal Medicine, School of Medicine, College of Medicine and Health Sciences, University of Gondar, Gondar, Ethiopia

* tseg729@gmail.com

**Data Availability Statement:** All relevant data are within the paper and its Supporting Information files.

## Abstract

### Introduction

Although the pathophysiological mechanism of hypertension is not fully elucidated yet, a large number of pieces of evidence have shown that genetic alterations in the renin-angio-tensin-aldosterone system play a central role. However, the association of *insertion/deletion* polymorphism of the angiotensin-converting enzyme (ACE) gene with essential hypertension is controversial yet, and there is a limited number of publications among the Ethiopian population. Therefore, this study aimed to determine the association of ACE gene I/D polymorphism with the risk of hypertension among essential hypertension patients at the University of Gondar Comprehensive Specialized Hospital, Gondar, Ethiopia.

### Materials and methods

A case-control study was conducted from October 07, 2020, to June 02, 2021, among hypertensive patients and normotensive control groups at the University of Gondar Comprehensive Specialized Hospital. A structured questionnaire was used to collect socio-demographic data and anthropometric measurements. Five milliliters of blood were drawn from each of the randomly selected 64 hypertensive and 64 normotensive participants for molecular test analysis. Genetic polymorphism of the ACE gene was identified using polymerase chain reaction (PCR) and electrophoresis. Data analysis was done using SPSS version 25.0 software. The strength of association between the genotype and hypertension was estimated through the calculation of adjusted odds ratio and 95% confidence intervals using logistic regression. P-value < 0.05 was considered statistically significant.

### Result

The distribution of DD genotypes and D allele of the ACE gene were 48.4% and 63% in essential hypertensive patients, respectively, while it were 29.7% and 42.2% in control

**Funding:** The author(s) received no specific funding for this work.

**Competing interests:** The authors have declared that no competing interests exist.

subjects respectively. The ACE DD genotype (p-value = 0.005) and D allele (p-value = 0.001) were more frequent among hypertensive patients as compared to controls.

## Conclusion

The present study found that the DD genotype and D allele of the ACE gene has had a strong association with a high risk of hypertension in the study population.

## 1. Introduction

Hypertension is a state of elevated blood pressure which can be defined as either systolic blood pressure at or above 130 mmHg and/or diastolic blood pressure at or above 80 mmHg [1]. Currently, hypertension is becoming one of the most common public health problems worldwide. Globally, it is estimated that around 40% of adults are living with hypertension with great regional and residence variations, and the majority of them (46%) are found in Africa [2]. Nearly one billion people have hypertension; of these, two-thirds are in developing countries [3]. Besides, in 2015, 8.5 million deaths were associated with high blood pressure, 88% of which were in low-income and middle-income countries [4]. The prevalence of high BP is predicted to increase by 24% in developed nations and by 80% in developing regions [5]. Likewise, the incidence of hypertension is increasing alarmingly in various populations of Ethiopia and other developing nations [2, 6, 7]. In Ethiopia, the estimate from meta-analysis and community-based studies revealed that the prevalence of hypertension ranged from 13% to 35%, and factors such as socio-demographic, economic, biological, and behavioral characteristics were found to be significantly associated with hypertension [2, 6–9]. However there are no genetic studies among Ethiopian hypertensive patients.

Numerous candidate genes have been implicated in susceptibility to essential hypertension [10]. In recent years, genes of the renin-angiotensin-aldosterone system (RAAS) have received a good deal of attention [11–13]. Thus, RAAS genes that encode for angiotensinogen (AGT), angiotensin type 1 receptor (AT1R) and angiotensin-converting enzyme (ACE) have been widely investigated in different ethnic populations [13]. The ACE gene that encodes ACE is one of them that has received a good deal of attention in recent studies. The key enzyme of this system (ACE) catalyzes the production of angiotensin II, which acts as a strong vasoobliterant and stimulates the secretion of aldosterone [10, 12, 14]. The ACE gene, which is 21 kb in size and has 26 exons, is found on chromosome 17q23. It has a polymorphism that results in the insertion or deletion of a 287 base pair Alu repetitive sequence in intron 16. Published studies have shown that the insertion/deletion (I/D) polymorphism in intron 16 of the ACE gene (ACE) accounts for approximately half the variance in ACE plasma levels [15, 16]. This might be owing to the existence of regulatory elements, such as a transcriptional silencer, in intron 16, which may affect gene transcription and result in increased gene expression if such a negative element is deleted [17]. The main function of this gene is the conversion of Angiotensin I to vasoactive, natriuretic octapeptide angiotensin II in the liver and inactivates a vasodilator peptide bradykinin [15, 18]. Consequently, the ACE gene has been postulated as a candidate gene for the development of essential HTN.

The (I/D) polymorphism of the ACE gene is shown to be associated with interpersonal variability and individuals carrying the deletion allele are associated with increased risk of HTN, but subsequent studies have yielded contrasting results [15, 16, 19]. Several studies conducted in Nigerian [20], Chinese [12, 13], Brazil [18], Indian [15], Arabian [21], and Bangladeshi [22]

populations have suggested that there is a strong association between ACE- DD polymorphism with a higher incidence of hypertension. On the contrary, other studies conducted in Gabon [23], Tunisian [24, 25], Spanish [26], India [16], Bangladesh [27], and Chinese [10] failed to show this association. Even one other study revealed an inverse association of them [28]. Patient selection, environmental factors, genetic allele combinations, and antihypertensive therapy may contribute to the inconsistent data from clinical studies on genetic polymorphisms associated with hypertension [13].

However, to the best of our knowledge, there are a limited number of studies on the issue in the study areas and study population. Therefore, the present study aimed to determine the association of ACE gene I/D polymorphism with the risk of hypertension among hypertensive patients with their corresponding control groups at the University of Gondar Comprehensive Specialized Hospital, Gondar, Ethiopia. The finding might result in an ACE gene genotype that may serve as a biomarker for early detection and diagnosis of HTN, and prevention of associated complications.

## 2. Materials and methods

### 2.1. Study design, setting, and period

A case-control study was conducted from October 07, 2020, to June 02, 2021, at the University of Gondar Comprehensive Specialized Hospital, Gondar, Ethiopia. The University of Gondar Comprehensive Specialized Hospital is one of the largest hospitals in the country. It is a tertiary teaching hospital serving a catchment area of approximately five million people. It has a follow-up clinic for major chronic illnesses including hypertension. The hypertension clinic is the one in which treatment and follow-up for hypertensive individuals is taking place.

### 2.2. Study participants

All essential hypertensive patients (as cases) and apparently healthy normotensive individuals (as controls) who visit the hospital were considered as the source population. All essential hypertensive patients who were newly diagnosed and at follow-up at the time of the study period were included as cases, while apparently healthy normotensive individuals attending the hospital during the study period were included as controls. For both cases and controls, individuals with age less than 18 years and greater than 65 years and/or who have clinically confirmed comorbidity such as tuberculosis, diabetes mellitus, liver disease, pregnancy hypertension, renal disease, inflammatory disease, thyroid disease, and all others with secondary hypertension were excluded. Pregnant or postpartum period women were also excluded from the study.

### 2.3. Sample size determination and sampling method

The sample size was calculated using G* power version 3.1.9.4 software by selecting an independent $t$-test [29]. It is calculated by considering alpha = 0.05, power (1- β) = 0.8 (80%), effect size (d) = 0.5 and allocation ratio $N_2/N_1$ = 1, then the total sample size became 128. Therefore, by a 1:1 case-control ratio of 64 hypertensive cases and 64 age sex-matched healthy controls were recruited as study participants using a simple random sampling technique.

### 2.4. Variables of the study

The dependent variable is systolic and diastolic BP, whereas ACE gene I/D polymorphism, socio-demographic factors (age, sex, education level, marital status, and residential area), behavioral factors (physical exercise, smoking, alcohol consumption, and salt intake),

anthropometric (BMI, waist to hip ratio), and family history of HTN was taken as explanatory variables.

## 2.5. Data collection procedure

The socio-demographic data were collected using semi-structured interviewer-administered questionnaires and medical record (chart) review (**Annex 3 in S1 File**). The questionnaires were prepared in English, then translated to the local language (Amharic), and re-translated back to English to check the consistency. They were developed based on the related literature of previously published papers and collected by experienced nurses. Body Mass Index (BMI) was also calculated from the body weight (kg) and height (meter) by measuring the weight of every study participant using a standard balance, and the height using a height measuring device attached to the balance. BMI = Weight (in kg) / (Height in m)$^2$. Using the WHO (2008) classification, five categories of BMI can be identified as follows: underweight, $<18.5$ kg/m$^2$; normal, 18.5–24.9 kg/m$^2$; overweight, 25.0–29.9 kg/m$^2$; and obesity, 30.0–39.9, extreme obesity $> 40.0$ kg/m$^2$.

Waist and hip circumference were measured using a standard non-stretchable measuring tape. The waist circumference was measured around the abdomen at the level midway (smallest horizontal girth) between the lowest rib margin and the iliac crest at the end of expiration. The hip circumference was also measured at the levels of widest diameters around the buttocks (at the broadest part of the hips). Then, waist to hip ratio (WHR) was calculated by dividing waist circumference by hip circumference. According to WHO (2019), the cut-off point considered for waist circumference (WC) was >88cm for females and >102cm for males to define overweight, the cut-off taken for waist to hip ratio was >0.85 for females and >0.90 for males to define overweight.

Blood pressure was measured after a minimum rest of 5 minutes, or 30 minutes for those who had drunk hot drinks such as coffee, with a sphygmomanometer at the midpoint of the left arm in the sitting position with arm support. The blood pressure was measured twice with an interval of 5 minutes, and the average value was taken as the true value. The cut-off points for elevated BP were SBP of 130mmHg or above and DBP of 80mmHg or above [1].

## 2.6. Blood samples collection and genetic analysis

The collection of blood samples was done by experienced medical laboratory technologists. A volume of 5ml of blood was collected using EDTA coated tube by certified health care professionals in the Hospital from each participant. Then, the blood samples were kept in a -4°C refrigerator for genetic analysis through the salting out method.

**2.6.1. Genomic DNA isolation.** DNA extraction was done using the non-enzymatic salting-out method [30], by taking 300μl of EDTA anti-coagulated blood of both patients and healthy controls and transferring it to a sterilized 1.5ml Eppendorf tube. The red blood cells (RBC) lysis buffer solution was used to lysis and remove RBCs. Similarly, white blood cells were lysed using a nuclear lysis buffer solution. Then, a highly concentrated salt (6M NaCl) was added to precipitate and remove proteins. The DNA was precipitated by chilling using isopropanol followed by washing with 70% ice-cold ethanol. Then, genomic DNA was dissolved with Tris-EDTA buffer (TE) and stored at -21°C till use. The quality of isolated genomic DNA was confirmed by using 1.5% agarose gel electrophoresis or by measuring its absorbance ratio at 260/280nm. Then, the samples were genotyped for ACE gene (I/D) polymorphism using sets of primers and appropriate PCR conditions (**Annexes 1 and 2 in S1 File**).

**2.6.2. Polymerase chain reaction (PCR).** The insertion/deletion (I/D) alleles of ACE gene polymorphisms were identified by TC 412 PCR thermocycler using specific primers. A

final volume (25 μl) of PCR reaction mixture was prepared using 10pmol of forward and reverse primers (forward primer 5′–CTGGAGACCACTCCCATCCTTTCT–3′ and reverse primer 5′GATGTGGCCATCACATTCGTCAGAT–3′) [15, 27, 30], 1.5 mM of MgCl$_2$, 0.2mM of each dNTP, 1.0 units of hot-start Taq polymerase, 2 μl of template DNA and water were used. The PCR amplification was set with an initial denaturation and activation of an enzyme at 95˚C for two minutes. Then, the DNA was amplified for 30 cycles. The cycle steps were denaturation at 94˚C for 30 seconds, annealing at 58˚C for 30 seconds and extension at 72˚C for 45 seconds followed by a final extension at 72˚C for 9 minutes. Finally, the PCR product was held at 4˚C until it becomes analyzed by agarose gel electrophoresis.

**2.6.3. Agarose gel electrophoresis.** PCR amplified products of *ACE I/D* genotypes were electrophoretically separated for 1 hour at 100 V on a 2% agarose gel. To stain and visualize DNA upon UV transillumination in gel, 1μl of 2% Ethidium Bromide was added. The PCR amplified products (12 μl) were mixed with 3 μl loading dye then loaded into wells of agarose gel. Electrophoresis was carried out in 1X tris acetate EDTA (TAE) buffer and the gel was visualized by a UV transilluminator. After electrophoresis, band sizes of 190bp (Deletion) and 490bp (Insertion) polymorphisms fragments were obtained and images were captured using the Gel-pro analyzer version 6.3 gel analysis software. Therefore, there were three genotypes after electrophoresis: A 490bp band (genotype II), a 190bp band (genotype DD), and both 490bp and 190bp band (genotype ID). The frequency of genotypes was calculated to determine the association of ACE I/D gene polymorphisms with hypertension and with other clinical characteristics.

## 2.7. Data processing and analysis

The data obtained from laboratory analyses of the blood samples and questionnaires were checked for completeness and cleaned by coding and entering the data into Epi-info statistical software version 7.0, and then exported into SPSS software version 25.0 package for analysis and further processing. Simple descriptive statistics were used to present the socio-demographic and clinical characteristics of the study subjects. The categorical variables were analyzed using the chi-square ($\chi^2$) test and are expressed as frequency and percentage. To determine the association of HTN with ACE gene polymorphism, we had analyzed it through a logistic regression model of analysis. The Hosmer and Lemeshow goodness-of-fit test was performed. Continuous variables are expressed as mean ± standard deviation (SD) and were analyzed using the independent t-test for comparing cases and control groups. They were also analyzed with variance (ANOVA) to identify the difference in the clinical characteristics of individuals with different ACE genotypes. The variables with $p < 0.05$ were declared statistically significant for all statistical tests.

## 2.8. Data quality assurance

To achieve quality, a two-day training was provided for data collectors. The training session includes the objective, procedure, risk, and benefit of the study. A pretest was carried out using 10% of the sample size at Gondar Poly Clinic before running the actual data collection. Then, corrections have been taken accordingly; the overall quality of laboratory analysis was maintained by strictly following the manufacturers' instructions and standard operational procedure (SOP) in the pre-analytic, analytic, and post-analytic stages of laboratory services. To catch any errors, the collected data were double-checked for completeness, consistency, accuracy, and clarity daily. The purity of extracted DNA was also checked by using a spectrophotometer of NanoDrop on which DNA yield at 260/280 nm ratio should be between 1.8 and 2.0. Repeated genotype was also done for about 10% of randomly selected samples as means of conformation.

## 2.9. Operational definitions

**2.9.1. Apparently healthy individuals (normotensives).**  Individuals with absence of disease based on clinical signs and symptoms and who have normal blood pressure (SBP <130 mm Hg or DBP <80 mm Hg) [1, 31].

**2.9.2. Hypertensive patients.**  Individuals who are clinically confirmed cases whose blood SBP ≥130 mm Hg or DBP ≥80 mm Hg) [1].

**2.9.3. High salt intake (Yes/No).**  If the participant up takes about 5grams or more (just more than a teaspoon) of sodium chloride salt (approximately 2gram sodium) per person per day, then he/she will be graded as a salt taker (Yes) [32].

**2.9.4. Regular physical exercise (Yes/No).**  A participant is classified as participating in regular physical exercise (Yes) if he/she does exercise three or more times per week and 30 or more minutes for each time [33].

**2.9.5. Current smokers (Yes/No).**  A participant is classified as a current smoker if he/she is currently smoking cigarettes, tobacco, bidis, or shisha daily [34].

**2.9.6. Current alcohol drinker (Yes/No).**  A participant is classified as a current alcohol drinker if he/she is consuming alcohol within the past one year [34].

## 2.10. Ethical considerations

This research was conducted after the ethical clearance was obtained from the Institutional Review Board (IRB) of University of Gondar (Ref. No: 2094/07/2020). Before starting data collection, an official permission letter was gained from the University of Gondar Comprehensive Specialized Hospital. Before starting the actual data collection process, written informed consent was acquired from each participant following the appropriate description of the study objective, benefits, and possible risks. Moreover, the confidentiality was assured by leaving out their name and telling the safety of the place where the data will be stored after collection.

## 3. Result

### 3.1. Socio-demographic and behavioral characteristics of study participants

A total of 128 (64 hypertensive patients and 64 controls) study participants were recruited in the study with a response rate of 100%. Of the total study participants, the majority 35 (54.7%) of cases and 33 (51.56%) of normotensive participants were females. The majority of the cases 38 (59.4%) were married. Most of the case groups, 52 (81.25%) and healthy controls 45 (70.31%) were non-alcohol drinkers; and 56 (87.5%) of the cases and 59 (92.19%) of healthy controls were nonsmokers. Most of the socio-demographic variables were not significantly associated with HTN. This implies these socio-demographic factors could not be taken as a reason for bias (**Table 1**).

### 3.2. Clinical and biochemical characteristics of the study participants

The mean ± SD age of cases and controls was 45.31 (±9.86) and 41.94 (±10.11) years, respectively. The mean BMI of cases and controls were 24.95 ± 5.19 kg/m$^2$ and 22.29 ± 3.46 kg/m$^2$, while their WHR was 0.91 ± 0.20 and 0.82 ± 0.09, respectively. The mean blood pressure of cases and controls was 132.50 (±11.20) / 82.03 (±8.58) and 107.19 (±9.34) / 71.41 (±8.89), respectively. The BMI, WHR, SBP, and DBP were significantly higher in hypertensive patients than in controls (p-value < 0.001). On the other hand, age (p-value = 0.058) was not significantly different between the study groups (**Table 2**).

**Table 1. Socio-demographic and behavioral characteristics of study participants at University of Gondar Comprehensive Specialized Hospital, Northwest Ethiopia, 2020/21 (n = 128).**

| Variables | | Study subjects | | | P-value |
|---|---|---|---|---|---|
| | | HTN (N = 64) | Control (N = 64) | Total (N = 128) | |
| Sex | Male | 29 (45.3%) | 31 (48.44%) | 60 (46.87%) | 0.723 |
| | Female | 35 (54.7%) | 33 (51.56%) | 68 (53.13%) | |
| Educational Status | Illiterate | 21 (32.8%) | 16 (25.0%) | 37 (28.9%) | 0.112 |
| | Primary education | 17 (26.6%) | 10 (15.6%) | 27 (21.1%) | |
| | Secondary education | 14 (21.9%) | 15 (23.4%) | 29 (22.6%) | |
| | Degree and above | 12 (18.7%) | 23 (36.0%) | 35 (27.4%) | |
| Marital status | Single | 12 (18.8%) | 20 (31.3%) | 32 (25.0%) | 0.208 |
| | Married | 38 (59.4%) | 31 (48.4%) | 69 (53.9%) | |
| | Divorced | 5 (7.8%) | 8 (12.5%) | 13 (10.2%) | |
| | Widowed | 9 (14.0%) | 5 (7.8%) | 14 (10.9%) | |
| Residential area | Urban | 49 (76.56%) | 54 (84.38%) | 103 (80.47%) | 0.265 |
| | Rural | 15 (23.44%) | 10 (15.62%) | 25 (19.53%) | |
| Current alcohol use | Yes | 12 (18.75%) | 19 (29.69%) | 31 (24.22%) | 0.149 |
| | No | 52 (81.25%) | 45 (70.31%) | 97 (75.78%) | |
| Current Smoking | Yes | 8 (12.5%) | 5 (7.81%) | 13 (10.15%) | 0.380 |
| | No | 56 (87.5%) | 59 (92.19%) | 115 (89.85%) | |
| Regular physical exercise | Yes | 26 (40.63%) | 30 (46.88%) | 56 (43.75%) | 0.476 |
| | No | 38 (59.37%) | 34 (53.12%) | 72 (56.25%) | |
| Salt intake | Yes | 39 (60.94%) | 45 (70.31%) | 84 (65.63%) | 0.264 |
| | No | 25 (39.06%) | 19 (29.69%) | 44 (34.37%) | |
| Family History of hypertension | Yes | 19 (29.69%) | 21 (32.81%) | 40 (31.25%) | 0.703 |
| | No | 45 (70.31%) | 43 (67.19%) | 88 (68.75%) | |

## 3.3. ACE genotype and allele distribution in HTN patients and healthy controls

The genomic DNA was isolated and its quality was confirmed by using agarose gel electrophoresis (**Fig 1**). Then, the qualified genomic DNA samples were amplified and genotyped through gel electrophoresis to identify the type of ACE gene polymorphism by using the size of base pairs of the bands formed. Then, the ACE gene was classed as II (490bp band), DD (190bp band), or ID (both 490bp and 190bp band) (**Fig 2**).

**Table 2. Clinical and biochemical characteristics of study participants at University of Gondar Comprehensive Specialized Hospital, Northwest Ethiopia, 2020/21 (n = 128).**

| Variables | Study subjects | | Sig. (2-tailed) |
|---|---|---|---|
| | HTN (N = 64) | Control (N = 64) | |
| | Mean±SD | Mean±SD | |
| Age (years) | 45.31±9.86 | 41.94±10.11 | 0.058 |
| BMI (kg/m$^2$) | 24.94±5.19 | 22.29±3.46 | 0.001* |
| WHR | 0.91±0.20 | 0.82±0.09 | 0.001* |
| SBP (mmHg) | 132.50±11.20 | 107.19±9.34 | 0.001* |
| DBP (mmHg) | 82.03±8.58 | 71.41±8.89 | 0.001* |

BMI = Body Mass Index, WHR = Waist-to-Hip Ratio, SBP = Systolic Blood Pressure, DBP = Diastolic Blood Pressure.

*statistically significant at p-value <0.05.

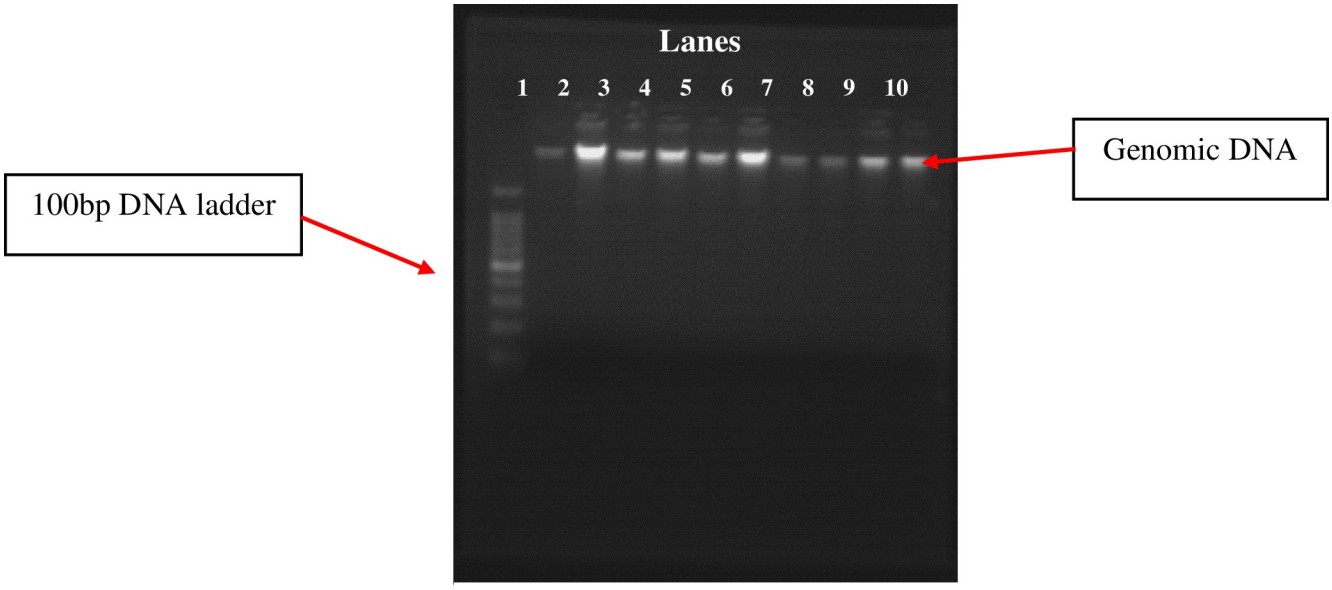

**Fig 1. Genomic DNA isolated through the salting-out method.**

The ACE genotypes frequency distribution in hypertensive patients and healthy control is given in **Table 3**. The distribution of II, ID, DD genotypes of the ACE gene was 21.9%, 29.7%, and 48.4% respectively in essential hypertensive patients and to 45.3%, 25.0%, and 29.7% in controls. The ACE DD genotype was more frequent in hypertensive patients (48.4%) as compared to controls (29.7%). However, ACE genotype II was less frequent in hypertensive

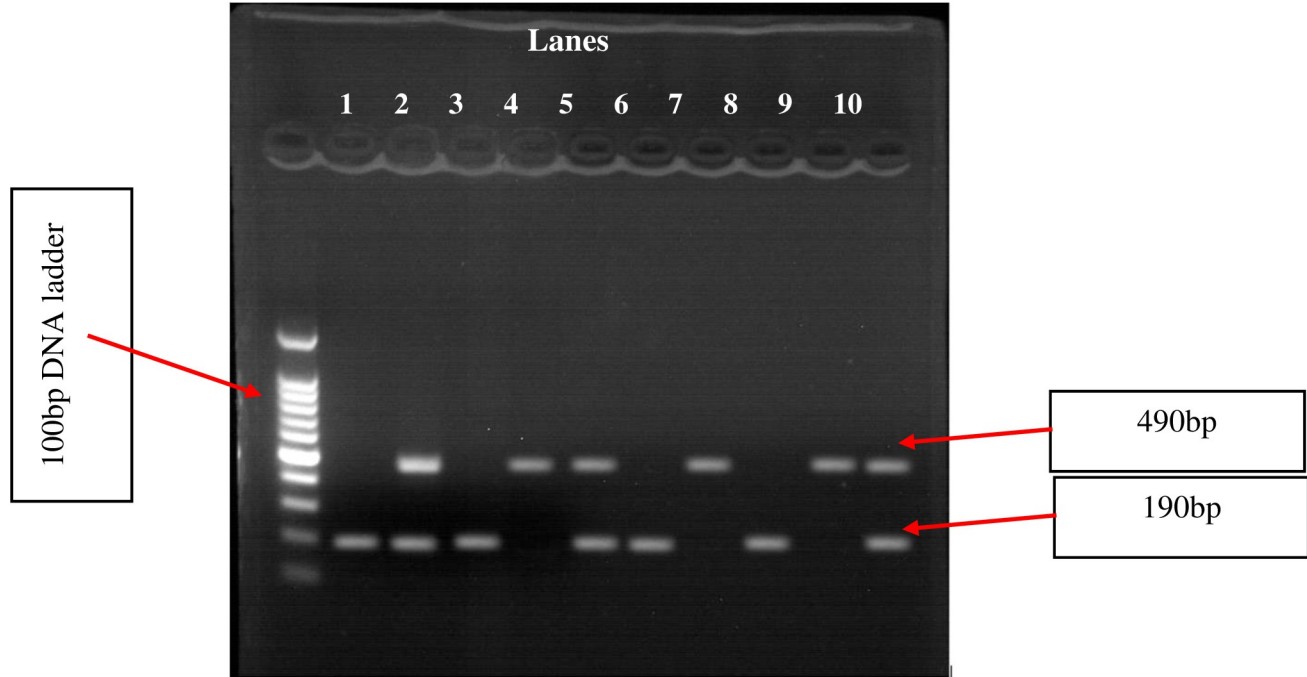

**Fig 2. Sample of gel electrophoresis result of ACE I/D gene polymorphism amplified with a specific pair of primers using conventional PCR.** (Lanes 1, 3, 6, and 8: homozygous DD genotypes; lanes 2, 5, and 10: heterozygous ID genotypes; lanes 4, 7, and 9: homozygous II genotypes).

**Table 3. Distribution of ACE genotypes and alleles frequency of ACE gene polymorphism (I/D) between HTN and controls (n = 128).**

| Genotype | HTN (n = 64) | Controls (n = 64) | Chi-squared | OR (95% CI) | P-value |
|---|---|---|---|---|---|
| DD | 31 (48.4%) | 19 (29.7%) | 8.02[a] | 3.38 (1.44,7.96) | 0.005 |
| ID | 19 (29.7%) | 16 (25.0%) | 3.73[b] | 2.46 (0.98, 6.18) | 0.056 |
| II | 14 (21.9%) | 29 (45.3%) | 0.00 | 1.00 (ref.) | |
| P-value[a] = 0.005, P-value[b] = 0.053 | | | | | |
| Allele frequencies | | | | | |
| D | 81 (63.3%) | 54 (42.2%) | 11.42[c] | 2.36 (1.43, 3.90) | 0.001 |
| I | 47 (36.7%) | 74 (57.8%) | 0.00 | 1.00 (ref.) | |
| | P-value[c] = 0.0007 | | | | |

Note:

[a, b, c] = the p-values of corresponding chi-squared values.

*P-value <0.05 considered highly significant.

patients (21.9%) in comparison to healthy controls (45.3%). The frequency of homozygous DD genotype in hypertensive patients was about three and a half fold higher than the healthy control group (OR, 3.38: 95% CI 1.44, 7.96). The allele frequency for the D allele is 0.63 in essential hypertension as compared to 0.42 in control subjects. D allele was about two times higher than the I allele in hypertensive patients (OR, 2.36; CI: 1.43, 3.90) with P-value < 0.001) compared to healthy controls (**Table 3**).

## 3.4. Comparison of the distribution of ACE I/D polymorphism in the present study population and different ethnic population

As it shown on the table below, the distribution of ACE I/D polymorphism in present study population and different ethnic population remains inconsistent (**Table 4**). As mentioned

**Table 4. Comparison of the distribution of ACE I/D polymorphism in present study population and different ethnic population.**

| Study population | HTN Cases | | | | Controls | | | | Sig. (2-tailed) | References |
|---|---|---|---|---|---|---|---|---|---|---|
| | N | ACE gene polymorphism (%) | | | N | ACE gene polymorphism (%) | | | | |
| | | DD | ID | II | | DD | ID | II | | |
| Ethiopian | 64 | 48.4 | 29.7 | 21.9 | 64 | 29.7 | 25.0 | 45.3 | * | Present study |
| Burkinabe | 202 | 66.83 | 28.22 | 4.95 | 204 | 35.78 | 50.98 | 13.24 | * | [35] |
| Gabonese | 95 | 57.9 | 34.7 | 7.4 | 37 | 40.5 | 51.4 | 8.1 | Ns | [23] |
| Egyptian | 110 | 39.9 | 53.6 | 15.5 | 93 | 26.9 | 55.9 | 17.2 | Ns | [5] |
| Tunisian | 388 | 44.6 | 45.4 | 10.0 | 425 | 48.2 | 42.4 | 9.4 | Ns | [25] |
| Nigerian | 612 | 45.3 | 42.8 | 11.9 | 612 | 38.4 | 49.5 | 12.1 | * | [20] |
| Bangladeshi | 44 | 50.0 | 38.6 | 11.4 | 59 | 23.7 | 44.1 | 33.2 | * | [22] |
| Spanish | 205 | 33.7 | 53.2 | 13.2 | 196 | 36.2 | 49.0 | 14.8 | Ns | [26] |
| Arabian | 235 | 41.7 | 57.4 | 0.9 | 333 | 40.8 | 48.9 | 10.3 | * | [21] |
| North Indian | 90 | 22.2 | 51.1 | 26.7 | 91 | 19.8 | 53.8 | 26.4 | Ns | [36] |
| South Indian | 208 | 38.9 | 32.7 | 28.4 | 220 | 20.0 | 26.4 | 53.6 | * | [15] |
| Chinese | 221 | 23.1 | 43.9 | 33.0 | 221 | 17.6 | 43.0 | 39.4 | * | [12] |
| Korean | 688 | 16.4 | 44.9 | 38.7 | 924 | 15.5 | 46.5 | 38.0 | Ns | [11] |
| Chile | 58 | 31.0 | 38.0 | 31.0 | 54 | 35.2 | 33.3 | 31.5 | * | [37] |

Note:

* = P-value <0.05, Ns = not significant.

earlier, the present study depicted that ACE I/D polymorphism was significantly associated with the risk of hypertension.

## 4. Discussion

Hypertension is one of the most common public health problems and is responsible for high cardiovascular morbidity and mortality worldwide [13]. The influence of ACE gene polymorphisms on hypertension is under controversy. Thus, we investigated the ACE gene polymorphism in Ethiopian hypertensive patients and the corresponding healthy controls.

The present study was the first one to be done in the Ethiopian population and revealed that the DD genotype of the ACE gene is strongly associated with the risk of hypertension. The DD genotype and D allele frequencies were found to be higher among the hypertensive patients (48.4% and 63.3%, respectively) than that of their corresponding controls (29.7% and 42.2%, respectively). The present finding also showed that individuals with the DD genotype of the ACE gene are about three times more likely to have hypertension than those with II genotype [OR = 3.38: 95% CI (1.44, 7.96)]. A strong association between the DD homozygous genotype and the D allele of the ACE I/D gene polymorphism with HTN was identified. It implies that the DD genotype and D allele might increase the susceptibility of getting hypertension among the participants and may also contribute to the relatively high prevalence of HTN in the Ethiopian population. This substantial association of the DD genotype and the D allele with increased risk of hypertension could be attributed to the deletion of a 287-bp non-coding region in the 16th intron of the ACE gene, which would result in increased ACE gene transcription and ACE activity. This might be related to the presence of a transcriptional silencer in intron 16, which result in increased gene expression and enzyme activity if such a suppressor fragment is deleted [17, 38]. The higher levels of ACE convert angiotensin-I to angiotensin-II, a potent vasoconstrictor; and also inactivate bradykinin, a potent vasodilator [15, 18]. High angiotensin-II levels can affect artery musculature, increase peripheral resistance, and raise blood pressure. It has also direct sodium-retaining effects through increasing the activity of the $Na^+/H^+$ exchanger and $Na^+/K^+$ ATPase in the proximal tubule, the $Na^+/K^+/2Cl^-$ transport in the loop of Henley, and multiple ion transporters in the distal nephron and collecting tubules. Angiotensin II also causes the production of aldosterone from the adrenal glands, which stimulates the epithelial cells of the kidneys to enhance salt and water reabsorption, resulting in increased blood volume and blood pressure, and hypertension [5, 26]. This substantial association of DD genotype with risk hypertension in the present finding indicates that individuals with DD genotype should have life style modification, do regular physical exercises, and minimize salt intake to reduce the synergetic effect of the genotype with the these factors to develop hypertension.

In agreement with the present study, many reports worldwide disclosed this association between ACE I/D polymorphism and HTN, i.e., in Nigerian [20], Burkina Faso [35], Chinese [12, 13], Brazil [18], South Indian [15], Chile [37], Arabian [21], and Bangladeshi [22] populations have suggested an association between ACE- DD polymorphism with a higher incidence of hypertension. In the contrast, many studies were conducted in Gabon [23], Tunisian [24, 25], Spanish [26], North Indian Punjabi [36], Bangladesh [27], Korean [11] and Chinese [10] population that failed to show this association. Some studies even showed an inverse association and show an inconsistent association [28]. The contradictory results regarding the involvement of the ACEI/D polymorphism in HTN are likely to be due to variance in ethnicity, heterogeneity of populations, geographical variation, sampling bias, and potentially other ecologic factors [15]. Furthermore, some environmental factors such as nutrition and physical activities have been associated with modification in epigenetic status [39]. Therefore, such

interaction between epigenetic modifications and polymorphisms describes the complexity of the genetic architecture that can underlie the inconsistency in association studies observed with the ACEI/D polymorphism in different populations [15, 18].

The BMI, WHR, SBP, and DBP were significantly higher in hypertensive patients than in controls (p-value < 0.001). The present study found that hypertensive patients had significantly higher BMI and WHR than normal controls. It might be because there is an association between obesity and higher blood pressure which may cause obesity-induced hypertension through the mechanism of insulin resistance, sodium retention, increased sympathetic nervous system activity, activation of renin-angiotensin-aldosterone, and altered vascular function [40, 41]. This finding was in line with the findings from Bangladesh [42], China [14], and South Korea [11], while it was inconsistent with the finding of a study conducted in southern India [15] and Egypt [5]. The inconsistency could be due to the variation in the ethnicity and socio-demographic condition of the study subjects. The present study found that hypertensive patients had significantly higher SBP and DBP than normal controls. This finding is in line with the study conducted in southern India [15], Bangladesh [42], China [14], and South Korea [11]. However, it is inconsistent with the study conducted in Egypt that reported there was no significant difference in SBP and DBP among cases and controls [43]. This discrepancy could be due to the very small sample size used by Egyptian studies and ethnic variation in study subjects.

### 4.1. Strengths and limitations

Selection bias was minimized by taking matched cases and controls in terms of sex and age variable, which are non-modifiable risk factors. The main limitation of our study was the limited sample size. Although the assay for serum ACE could not be performed, such tests might have provided a more accurate assessment of the relationship between ACE genotype and blood pressure.

### 5. Conclusion

The current study discovered that the ACE I/D polymorphism is associated with the risk of HTN. The ACE gene DD genotype and D allele predisposes to the occurrence of hypertension in our study participants. As a result, the ACE gene I/D polymorphism might be used as a biomarker for early diagnosis and detection of hypertension and prevention of its complication. To utilize the DD genotype as a marker for hypertension risk, substantial sample size studies on the same and diverse populations are required. Further studies with a larger sample size are needed to better understand the possible role of ACE gene I/D polymorphism with essential HTN in the Ethiopian population.

### Supporting information

**S1 Data.**
(XLSX)

**S1 File.**
(DOCX)

**S1 Raw images.**
(DOCX)

## Acknowledgments

The authors would like to acknowledge the Department of Biotechnology, the University of Gondar for providing laboratory equipment and facilities for conducting the molecular analysis.

## Author Contributions

**Conceptualization:** Tsegaye Adane Birhan, Meseret Derbew Molla, Kibur Hunie Tesfa.

**Data curation:** Tsegaye Adane Birhan.

**Formal analysis:** Tsegaye Adane Birhan.

**Investigation:** Tsegaye Adane Birhan.

**Methodology:** Tsegaye Adane Birhan, Meseret Derbew Molla, Mohamed Abdulkadir, Kibur Hunie Tesfa.

**Project administration:** Tsegaye Adane Birhan.

**Resources:** Tsegaye Adane Birhan.

**Software:** Tsegaye Adane Birhan.

**Supervision:** Meseret Derbew Molla, Mohamed Abdulkadir, Kibur Hunie Tesfa.

**Validation:** Meseret Derbew Molla, Mohamed Abdulkadir, Kibur Hunie Tesfa.

**Writing – original draft:** Tsegaye Adane Birhan.

**Writing – review & editing:** Meseret Derbew Molla, Mohamed Abdulkadir, Kibur Hunie Tesfa.

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
