## [Decision Letter · Decision Letter 0]

31 May 2022

PONE-D-22-09370Association of angiotensin-converting enzyme gene insertion/deletion polymorphisms with risk of hypertension among the Ethiopian populationPLOS ONE

Dear Dr. Birhan,

Thank you for submitting your manuscript to PLOS ONE. After careful consideration, we feel that it has merit but does not fully meet PLOS ONE’s publication criteria as it currently stands. Therefore, we invite you to submit a revised version of the manuscript that addresses the points raised during the review process.

We look forward to receiving your revised manuscript.

Kind regards,

Kanhaiya Singh, Ph.D

Academic Editor

PLOS ONE

Journal Requirements:

5. Please ensure that you refer to Figure 1 in your text as, if accepted, production will need this reference to link the reader to the figure.

6. Please upload a copy of Figure 3 and 4, to which you refer in your text on page 12. If the figure is no longer to be included as part of the submission please remove all reference to it within the text.

Additional Editor Comments:

Although the reviewers have found merit in this work, they have raised concerns about the statistical interpretation of the results. They have also recommended major edits in the text of the manuscript.

Reviewers' comments:

Reviewer's Responses to Questions

**Comments to the Author**

1. Is the manuscript technically sound, and do the data support the conclusions?

Reviewer #1: Partly

Reviewer #2: Yes

Reviewer #3: Yes

Reviewer #4: Yes

2. Has the statistical analysis been performed appropriately and rigorously? 

Reviewer #1: No

Reviewer #2: Yes

Reviewer #3: Yes

Reviewer #4: Yes

3. Have the authors made all data underlying the findings in their manuscript fully available?

Reviewer #1: Yes

Reviewer #2: Yes

Reviewer #3: Yes

Reviewer #4: Yes

4. Is the manuscript presented in an intelligible fashion and written in standard English?

Reviewer #1: Yes

Reviewer #2: Yes

Reviewer #3: Yes

Reviewer #4: Yes

5. Review Comments to the Author

Reviewer #1: This manuscript presents interesting results, however there are some major flaws with the study. It needs to be proof read and some sections need to be written more concisely. Introduction and discussion portion need to be written strongly. Statistical analysis also has some flaws. Kindly check the attached file for more detailed comments.

Reviewer #2: The manuscript “PONE-D-22-09370” is aimed to find the association of ACE gene insertion/deletion polymorphisms with risk of hypertension among the Ethiopian population. Manuscript is interesting and discussed properly. Some minor changes could attract more audience and more justified.

1. Angiotensin-converting enzyme (ACE) gene insertion/deletion (I/D) polymorphism were reported as a risk factor for other populations as discussed in introduction part of paper, a systemic meta-analysis results would make manuscript interesting.

2. Comparison of genotype of ACE gene insertion/deletion (I/D) with ACE serum level would make results more exciting.

Reviewer #3: The manuscript NO. PONE-D-22-09370 is well written and very important for the field, therefore, I recommend the manuscript for publishing but after improving the grammatical and typing error in the manuscript.

Reviewer #4: This is an association study angiotensin-converting enzyme gene insertion/deletion polymorphisms with risk of hypertension among the Ethiopian population. The study included 64 hypertensive patients and 64 controls. While the data is interesting, the manuscript has significant omissions. My primary concerns are the following:

- The finding is interesting and involves an important public health issue but the key problem with this study is that in the complex phenotype field, few researchers would have confidence in an association that is based on these small sample sizes. (Even if the authors stated as the power of the study).

- The discussion could be made much more interesting. As it is, they primarily reiterate their results and provide a bit more literature review. However, it would be interesting if they could discuss clinical implications and areas for further study.

- Authors have listed ACE I/D polymorphism and HTNI reports in different population. I would appreciate if author will provide a table with frequency and association results from different population. It will help readers to have a glance at global variation for the polymorphism association HTN1.

6. PLOS authors have the option to publish the peer review history of their article (what does this mean?). If published, this will include your full peer review and any attached files.

Reviewer #1: No

Reviewer #2: **Yes: **PRIYANKA VEMRA

Reviewer #3: No

Reviewer #4: No

---

## [Author Response · Author response to Decision Letter 0]

21 Jul 2022

Title: Association of angiotensin-converting enzyme gene insertion/deletion polymorphisms with risk of hypertension among the Ethiopian population

Manuscript ID: PONE-D-22-09370

Point by point response to editor and reviewers comment 

Dear Editor/reviewers, we are very grateful for your effective comments given to our manuscript. We have considered your eminent comments and suggestions and modified the manuscript accordingly. The responses are highlighted in the main manuscript.

Authors’ response: Dear reviewer, thank you very much for your essential comments for the improvement of the manuscript. 

The comments are addressed and revised point by point as follows:

Review Comments to the Author

Reviewer #1: This manuscript presents interesting results, however there are some major flaws with the study. It needs to be proof read and some sections need to be written more concisely. Introduction and discussion portion need to be written strongly. Statistical analysis also has some flaws. Kindly check the attached file for more detailed comments.

Response for Reviewer #1: Thank you very much. We have done the proof reading accordingly to have well written manuscript. Some literatures are added in the introduction and discussion section to make the paper stronger. The statistical analysis section have been modified accordingly (Page 10&11, Table 1 and Page 11&12, Table 2).

Reviewer #2: The manuscript “PONE-D-22-09370” is aimed to find the association of ACE gene insertion/deletion polymorphisms with risk of hypertension among the Ethiopian population. Manuscript is interesting and discussed properly. Some minor changes could attract more audience and more justified.

1. Angiotensin-converting enzyme (ACE) gene insertion/deletion (I/D) polymorphism were reported as a risk factor for other populations as discussed in introduction part of paper, a systemic meta-analysis results would make manuscript interesting.

2. Comparison of genotype of ACE gene insertion/deletion (I/D) with ACE serum level would make results more exciting.

Response for Reviewer #2: 

1. Yes of course, the ACE gene I/D polymorphism as a risk factor of HTN is studied in different population. But still no study in the current population specifically. And the Meta analysis will be done after the publication of this finding. 

2. Really, the comparison of genotype of ACE gene insertion/deletion (I/D) with ACE serum level would give better result and finding. However, unluckily we have no access to the reagents for enzymatic assay and it is already mentioned in the limitation section of the manuscript.

Reviewer #3: The manuscript NO. PONE-D-22-09370 is well written and very important for the field, therefore, I recommend the manuscript for publishing but after improving the grammatical and typing error in the manuscript.

Response to Reviewer #3: Thank you. We have tried to improve the grammatical and typing error in the manuscript accordingly.

Reviewer #4: This is an association study angiotensin-converting enzyme gene insertion/deletion polymorphisms with risk of hypertension among the Ethiopian population. The study included 64 hypertensive patients and 64 controls. While the data is interesting, the manuscript has significant omissions. My primary concerns are the following:

a) The finding is interesting and involves an important public health issue but the key problem with this study is that in the complex phenotype field, few researchers would have confidence in an association that is based on these small sample sizes. (Even if the authors stated as the power of the study).

b) The discussion could be made much more interesting. As it is, they primarily reiterate their results and provide a bit more literature review. However, it would be interesting if they could discuss clinical implications and areas for further study.

c) Authors have listed ACE I/D polymorphism and HTNI reports in different population. I would appreciate if author will provide a table with frequency and association results from different population. It will help readers to have a glance at global variation for the polymorphism association HTN1

Response to Reviewer #4: Thank you very much for your constructive comments, we fully agreed with the comments and have been justified the comments below.

a) Even though the sample size was calculated statistically, with this small sample size generalization would be difficult. We have also included it as a limitation and suggested as recommendation better to be done with larger sample size. But still since there is no study conducted in the study area, this finding would be utilized as a baseline data to conduct further advanced studies with lager sample size. The main reason to have small sample size in this study is due to the financial issues and has been recommended for further researchers in the main body of the manuscript (Page 18, line 381-383).

b) We have included some clinical implications and areas for further study in the discussion section in order to improve the quality of the paper (Page 16, line 333-337).

c) We have attempted to include a table elaborating the distribution of ACE I/D polymorphism in different population and compared it with the present finding (Page 14, line 291-296 and Page 15, Table 4).

---

## [Decision Letter · Decision Letter 1]

15 Aug 2022

PONE-D-22-09370R1Association of angiotensin-converting enzyme gene insertion/deletion polymorphisms with risk of hypertension among the Ethiopian populationPLOS ONE

Dear Dr. Birhan,

Thank you for submitting your manuscript to PLOS ONE. After careful consideration, we feel that it has merit but does not fully meet PLOS ONE’s publication criteria as it currently stands. Therefore, we invite you to submit a revised version of the manuscript that addresses the points raised during the review process.

We look forward to receiving your revised manuscript.

Kind regards,

Kanhaiya Singh, Ph.D

Academic Editor

PLOS ONE

Journal Requirements:

Additional Editor Comments (if provided):

Please address to the comments raised by Reviewer 1

Reviewers' comments:

Reviewer's Responses to Questions

**Comments to the Author**

1. If the authors have adequately addressed your comments raised in a previous round of review and you feel that this manuscript is now acceptable for publication, you may indicate that here to bypass the “Comments to the Author” section, enter your conflict of interest statement in the “Confidential to Editor” section, and submit your "Accept" recommendation.

Reviewer #1: (No Response)

Reviewer #2: All comments have been addressed

Reviewer #4: All comments have been addressed

2. Is the manuscript technically sound, and do the data support the conclusions?

Reviewer #1: Yes

Reviewer #2: Yes

Reviewer #4: Yes

3. Has the statistical analysis been performed appropriately and rigorously? 

Reviewer #1: Yes

Reviewer #2: Yes

Reviewer #4: Yes

4. Have the authors made all data underlying the findings in their manuscript fully available?

Reviewer #1: Yes

Reviewer #2: Yes

Reviewer #4: Yes

5. Is the manuscript presented in an intelligible fashion and written in standard English?

Reviewer #1: Yes

Reviewer #2: Yes

Reviewer #4: Yes

6. Review Comments to the Author

Reviewer #1: Authors have included corrections and justified almost all the raised concerns. However, there are few more concerns such as:

1. The primer sequence given to amplify the ACE gene polymorphism gives a PCR product of 191bp and the insertion sequence is 287bp, which makes it 478bp (in case of insertion). Then why 490bp is mentioned in the text?

2. Page 13, line 277: Figure 2 should not be mentioned as it is only representative figure showing different genotypes, not the frequency distribution.

3. Table 3 legend: "Distribution of ACE genotypes and alleles frequency of ACE gene polymorphism (I/D) between HTN and controls at University of Gondar Comprehensive Specialized Hospital, Northwest Ethiopia, 2020/21 (n=128)". There is no need to mention the university name and location in table heading.

4. In Table 3, what does a, b, and c correspond to in various values as well as in p-values?

5. Three Un-cropped gel images for isolated DNA samples have been provided by the authors in supplementary file. However all the three images are same, with different angles and color contrast. And the other two images provided for amplified PCR products, One image is the replicate of representative gel picture in the paper given earlier and the other image is a very bad one for doing genotyping. If these images are the best representative ones then looking at these images raises the question about the technical precision and methods used for PCR as well as gel electrophoresis,

Reviewer #2: Authors have successfully addressed my concerns. I suggest to accept the manuscript in its current version.

Reviewer #4: Thank you for the response. The authors have addressed my concerns.

7. PLOS authors have the option to publish the peer review history of their article (what does this mean?). If published, this will include your full peer review and any attached files.

Reviewer #1: No

Reviewer #2: No

Reviewer #4: No

---

## [Author Response · Author response to Decision Letter 1]

27 Aug 2022

Title: Association of angiotensin-converting enzyme gene insertion/deletion polymorphisms with risk of hypertension among the Ethiopian population

Manuscript ID: PONE-D-22-09370R1

Point by point response to editor and reviewers comment 

Dear Editor/reviewers, we are very grateful for your effective comments given to our manuscript. We have considered your eminent comments and suggestions and modified the manuscript accordingly. The responses are highlighted in the main manuscript. The comments are addressed and revised point by point as follows.

Review comments to the Author

Reviewer #1: Authors have included corrections and justified almost all the raised concerns. However, there are few more concerns such as:

1. The primer sequence given to amplify the ACE gene polymorphism gives a PCR product of 191bp and the insertion sequence is 287bp, which makes it 478bp (in case of insertion). Then why 490bp is mentioned in the text?

2. Page 13, line 277: Figure 2 should not be mentioned as it is only representative figure showing different genotypes, not the frequency distribution.

3. Table 3 legend: "Distribution of ACE genotypes and alleles frequency of ACE gene polymorphism (I/D) between HTN and controls at University of Gondar Comprehensive Specialized Hospital, Northwest Ethiopia, 2020/21 (n=128)". There is no need to mention the university name and location in table heading.

4. In Table 3, what does a, b, and c correspond to in various values as well as in p-values?

5. Three Un-cropped gel images for isolated DNA samples have been provided by the authors in supplementary file. However all the three images are same, with different angles and color contrast. And the other two images provided for amplified PCR products, one image is the replicate of representative gel picture in the paper given earlier and the other image is a very bad one for doing genotyping. If these images are the best representative ones then looking at these images raises the question about the technical precision and methods used for PCR as well as gel electrophoresis.

Response for Reviewer #1: Thank you very much for your constructive comments. We fully agree with the comments and have justified the comments below. 

1. The insertion fragment of PCR product seems to be about 487bp through the mathematical addition of the 190bp and the deleted alu of 287bp. However, previously established method for PCR detection of the insertion/deletion polymorphism of the human ACE gene has determined the fragment and found it to be 490bp (Rigat et al., 1992). It is also supported by several studies (Dai et al., 2019, Aggarwal et al., 2017, Pavlyushchik et al., 2016, Krishnan et al., 2016, Alsafar et al., 2015, Zawilla et al., 2014, Rashed et al., 2015). Moreover, in the present study the band is formed at about 490bp of the reference ladder. That is why we included it in the manuscript's text. 

2. Thank you; we agree fully. We have removed it from the text and only Table 3 is mentioned to represent the frequency distribution showing different genotypes. (Page 13, Line 277). 

3. On the legend of the Table 3, we have included the hospital name "University of Gondar Comprehensive Specialized Hospital" to have an informative title. We have removed it if it is deemed irrelevant. (Page 14, Line 288). 

4. In Table 3, the superscript a, b, and c labeled on the chi-squared values were used to show the corresponding p-values leveled with corresponding letters (a, b, and c). We reasoned that mentioning the p-values of odds ratio and chi-square in the same table would be confusing to the readers. Therefore, it’s better to use letter labels to allocate the p-values of each chi-squared value. (Page 14, Line 289). 

5. Yes, of course, there was duplication. Therefore, we have included a single un-cropped gel image for isolated DNA samples and a single gel image of amplified PCR products. Regarding the quality of the gel, of course, these images were not the best representatives of the gel runs we have recorded. However, we didn’t store the images captured during experimental runs and simply included them as it is what is at our hand. (S1_raw_images) Moreover, the images included in the manuscript are of better quality, which can resolve the quality concerns.

References:

AGGARWAL, N., KARE, P. K., VARSHNEY, P., KALRA, O. P., MADHU, S. V., BANERJEE, B. D., YADAV, A., RAIZADA, A. & TRIPATHI, A. K. 2017. Role of angiotensin converting enzyme and angiotensinogen gene polymorphisms in angiotensin converting enzyme inhibitor-mediated antiproteinuric action in type 2 diabetic nephropathy patients. World journal of diabetes, 8, 112-119.

ALSAFAR, H., HASSOUN, A., ALMAZROUEI, S., KAMAL, W., ALMAINI, M., ODAMA, U. & RAIS, N. 2015. Association of Angiotensin Converting Enzyme Insertion-Deletion Polymorphism with Hypertension in Emiratis with Type 2 Diabetes Mellitus and Its Interaction with Obesity Status. Disease Markers, 2015, 7.

DAI, S., DING, M., LIANG, N., LI, Z., LI, D., GUAN, L. & LIU, H. 2019. Associations of ACE I/D polymorphism with the levels of ACE, kallikrein, angiotensin II and interleukin-6 in STEMI patients. Scientific Reports, 9, 19719.

KRISHNAN, R., SEKAR, D., KARUNANITHY, S. & SUBRAMANIUM, S. 2016. Association of angiotensin converting enzyme gene insertion/deletion polymorphism with essential hypertension in south Indian population. Genes & Diseases, 3, 159-163.

PAVLYUSHCHIK, O. O., AFONIN, V. Y., SAROKINA, V. N., CHAK, T. A., KHAPALIUK, A. V. & ANISOVICH, M. V. 2016. Association of the ACE I/D gene polymorphism with DNA damage in hypertensive men. Cytology and Genetics, 50, 304-311.

RASHED, L., ABDEL HAY, R., MAHMOUD, R., HASAN, N., ZAHRA, A. & FAYEZ, S. 2015. Association of Angiotensin-Converting Enzyme (ACE) Gene Polymorphism with Inflammation and Cellular Cytotoxicity in Vitiligo Patients. PLOS ONE, 10, e0132915.

RIGAT, B., HUBERT, C., CORVOL, P. & SOUBRIER, F. 1992. PCR detection of the insertion/deletion polymorphism of the human angiotensin converting enzyme gene (DCP1) (dipeptidyl carboxypeptidase 1). Nucleic Acids Res, 20, 1433.

ZAWILLA, N., SHAKER, D., ABDELAAL, A. & AREF, W. 2014. Angiotensin-converting enzyme gene polymorphisms and hypertension in occupational noise exposure in Egypt. International journal of occupational and environmental health, 20, 194-206.

---

## [Decision Letter · Decision Letter 2]

14 Sep 2022

PONE-D-22-09370R2Association of angiotensin-converting enzyme gene insertion/deletion polymorphisms with risk of hypertension among the Ethiopian populationPLOS ONE

Dear Dr. Birhan,

Thank you for submitting your manuscript to PLOS ONE. After careful consideration, we feel that it has merit but does not fully meet PLOS ONE’s publication criteria as it currently stands. Therefore, we invite you to submit a revised version of the manuscript that addresses the points raised during the review process.

We look forward to receiving your revised manuscript.

Kind regards,

Kanhaiya Singh, Ph.D

Academic Editor

PLOS ONE

Additional Editor Comments:

As pointed by reviewer 1, please provide raw gel images related to PCR genotyping for all the samples genotyped.

Reviewers' comments:

Reviewer's Responses to Questions

**Comments to the Author**

1. If the authors have adequately addressed your comments raised in a previous round of review and you feel that this manuscript is now acceptable for publication, you may indicate that here to bypass the “Comments to the Author” section, enter your conflict of interest statement in the “Confidential to Editor” section, and submit your "Accept" recommendation.

Reviewer #1: (No Response)

2. Is the manuscript technically sound, and do the data support the conclusions?

Reviewer #1: Partly

3. Has the statistical analysis been performed appropriately and rigorously? 

Reviewer #1: Yes

4. Have the authors made all data underlying the findings in their manuscript fully available?

Reviewer #1: No

5. Is the manuscript presented in an intelligible fashion and written in standard English?

Reviewer #1: Yes

6. Review Comments to the Author

Reviewer #1: The authors have answered the queries. However, I believe the authors should provide all the raw gel images related to PCR genotyping (for all the samples). It would help to resolve the question of data authenticity.

7. PLOS authors have the option to publish the peer review history of their article (what does this mean?). If published, this will include your full peer review and any attached files.

Reviewer #1: No

---

## [Author Response · Author response to Decision Letter 2]

22 Sep 2022

Title: Association of angiotensin-converting enzyme gene insertion/deletion polymorphisms with risk of hypertension among the Ethiopian population

Manuscript ID: PONE-D-22-09370R2

A point-by-point response to the editor and reviewer's comment 

Dear Editor/reviewers, we are very grateful for your effective comments given to our manuscript. We have considered your eminent comments and suggestions and modified the manuscript accordingly. The responses are highlighted in the main manuscript. The comments are addressed and revised point by point as follows.

Response to the Editor:

Since all the data related to genotype (II, ID, or DD) for all the samples are included in the supplementary file, simply we have taken sample/representative gel images to show how the ACE gene I/D polymorphism of all samples was identified. Moreover, previously published articles, including the one published in this journal (Rashed et al., 2015), have taken representative gel images only and simply included the genotype information of all samples in the tables of supplementary documents. We did the same thing and don't believe it is mandatory to include the gel pictures for all 128 samples because the genotypes of all 128 samples are contained in the supplementary file.

Review comments to the Author

Reviewer #1: The authors have answered the queries. However, I believe the authors should provide all the raw gel images related to PCR genotyping (for all the samples). It would help to resolve the question of data authenticity.

Response for Reviewer #1: Thank you very much for your constructive comments. We fully agree with the comments and have justified the comments below. 

1. All the data are fully available within the supplementary document including the ACE gene I/D polymorphism of all 128 samples. However, we have taken representative photographs/images of gel images. Having this, since we have considered the representative gel images/photographs enough, we didn’t capture the images of 128 samples. Simply we have recorded and entered the gene polymorphism data after observing the formation of bands as it is whether at 190bp, 490bp, or on both 190bp and490bp. Simply we have taken a sample/representative gel image to show how the ACE gene I/D polymorphism of all samples was identified. But the types of ACE gene I/D polymorphism of all samples could be obtained in the supplementary document attached to the journal. Therefore, the presence of all the raw gel images for all the samples may not be mandatory and could not significantly affect the quality of the paper. 

Furthermore, previously published articles, including those published in PLOS ONE (Rashed et al., 2015), simply have included a sample of gel image of PCR amplified products. They didn’t include the gel images of all the samples they performed and they have included the genotypes in the supplementary document. 

Reference:

RASHED, L., ABDEL HAY, R., MAHMOUD, R., HASAN, N., ZAHRA, A. & FAYEZ, S. 2015. Association of Angiotensin-Converting Enzyme (ACE) Gene Polymorphism with Inflammation and Cellular Cytotoxicity in Vitiligo Patients. PLOS ONE, 10, e0132915.

---

## [Editor Report · Decision Letter 3]

27 Sep 2022

Association of angiotensin-converting enzyme gene insertion/deletion polymorphisms with risk of hypertension among the Ethiopian population

PONE-D-22-09370R3

Dear Dr. Birhan,

We’re pleased to inform you that your manuscript has been judged scientifically suitable for publication and will be formally accepted for publication once it meets all outstanding technical requirements.

Kind regards,

Kanhaiya Singh, Ph.D

Academic Editor

PLOS ONE